# *Drosophila* Larval Models of Invasive Tumorigenesis for In Vivo Studies on Tumour/Peripheral Host Tissue Interactions during Cancer Cachexia

**DOI:** 10.3390/ijms22158317

**Published:** 2021-08-02

**Authors:** Joseph A. Hodgson, Jean-Philippe Parvy, Yachuan Yu, Marcos Vidal, Julia B. Cordero

**Affiliations:** 1CRUK Beatson Institute, Institute of Cancer Sciences, University of Glasgow, Garscube Estate, Switchback Road, Glasgow G61 1BD, UK; joseph.hodgson501@gmail.com (J.A.H.); jeanfix@gmail.com (J.-P.P.); y.yu@beatson.gla.ac.uk (Y.Y.); 2Wolfson Wohl Cancer Research Centre, Institute of Cancer Sciences, University of Glasgow, Garscube Estate, Switchback Road, Glasgow G61 1QH, UK

**Keywords:** *Drosophila* larvae, *Ras*, *Scribble*, dual driver system, cancer cachexia

## Abstract

Cancer cachexia is a common deleterious paraneoplastic syndrome that represents an area of unmet clinical need, partly due to its poorly understood aetiology and complex multifactorial nature. We have interrogated multiple genetically defined larval *Drosophila* models of tumourigenesis against key features of human cancer cachexia. Our results indicate that cachectic tissue wasting is dependent on the genetic characteristics of the tumour and demonstrate that host malnutrition or tumour burden are not sufficient to drive wasting. We show that JAK/STAT and TNF-α/Egr signalling are elevated in cachectic muscle and promote tissue wasting. Furthermore, we introduce a dual driver system that allows independent genetic manipulation of tumour and host skeletal muscle. Overall, we present a novel *Drosophila* larval paradigm to study tumour/host tissue crosstalk in vivo, which may contribute to future research in cancer cachexia and impact the design of therapeutic approaches for this pathology.

## 1. Introduction

Cancer is one of the leading causes of death worldwide. Even though the devastating effects of cancer are often linked to non-tissue-autonomous consequences of tumour burden, the understanding of the interactions between tumours and host tissues remains incomplete. One such outcome of these interactions is the onset of cancer cachexia, a paraneoplastic syndrome defined as the “loss of skeletal muscle mass (with or without loss of fat mass) that cannot be fully reversed by conventional nutritional support and leads to progressive functional impairment” [1].

Cachexia is a multifactorial and multi-organ pathology, which is thought to cause up to 30% of late-stage cancer-related deaths [2,3]. Clinical manifestations of cachexia include asthenia, anorexia, significant loss of body fat and muscle, metabolic deregulation, abdominal fluid accumulation, systemic inflammation and immune infiltration of various tissues [4,5]. Cachexia is identified in up to 80% of advanced-stage cancer patients [6] and reduces tolerance to treatment, therapeutic response, patient quality of life and survival [7,8]. Nevertheless, there is no clear gold standard therapy for cachexia, and it therefore represents a major unmet clinical need [9].

One of the problems at the root of the dearth of therapeutic interventions for the treatment of cachexia is the fact its aetiology is poorly understood. Cachexia is a multifactorial condition further complicated by the heterogeneity of the general population, tumours themselves and the therapies used to treat them. This is evidenced by observations that patients with apparently similar tumours can have very different responses in terms of developing cachexia [10] and the fact cachexia is most frequently associated with various metastatic tumour types, such as pancreatic, lung, gastric and renal-cell cancers [11,12,13]. These characteristics are maintained in mouse models, where histologically similar tumours result in distinct cachectic outcomes [14], and in tumour cell lines from different sources, which have widely variable effects [15]. Given the predisposition of certain tumour types for the induction of cachexia, there are likely to be one or more molecular mechanisms that underpin whether a tumour will be cachectogenic, yet the heterogeneity and complexity of higher models makes the identification of pro-wasting mechanisms very difficult.

*Drosophila* has been successfully utilised as a model system to understand the growth and development of cancer [16,17,18] and to investigate the communication between host tissues and the tumour itself [19,20,21,22]. Several adult *Drosophila* tumour models have been used to identify tumour-secreted factors contributing to peripheric tissue wasting [23,24], making tumours refractory to the action of pro-wasting factors [25] and driving host anorexia [26]. A *Drosophila* larval model of high-sugar diet (HSD)-enhanced tumourigenesis identified important metabolically induced molecular changes within tumours driving muscle wasting [27]. While these studies have provided invaluable insight into tumour driven mechanisms of cancer cachexia, less is known about the molecular changes occurring within peripheric tissues and their functional role in tumour-driven host tissue wasting.

Here, we analyse several genetically defined larval *Drosophila* tumour models and show that only some of them recapitulate molecular and phenotypic aspects of cancer cachexia. We closely characterize the aetiology of the process and define primary versus secondary factors contributing to cachexia. Our results unambiguously point to the genetic characteristics of the tumour, rather than tumour burden or anorexia, as key primary determinants in the development of cachexia. We analysed transcriptional changes in wasting muscles and identify molecular pathways involved in wasting. Importantly, we establish a novel *Drosophila* larval model of cancer cachexia using a dual-driver system that allows parallel and independent genetic manipulation of muscle and tumour and which would facilitate studies of host-tumour interactions in cachexia and beyond.

## 2. Results and Discussion

### 2.1. Cachexia Is Dependent on Tumour Genotype

Loss of function mutations in cell polarity genes can induce neoplastic tumour formation in larval imaginal discs. These transformed neoplastic epithelial discs result in larval developmental delay and eventual death [16,28,29]. Expression of oncogenic *Ras^V12^* in these neoplastic tumours creates more aggressive invasive tumours that are able to invade distal tissues [18,28]. Hyperplastic tumours can also be generated in *Drosophila* by hyperactivation of the Hippo pathway [24,29]. As in human cancer patients, adult *Drosophila* tumour models have revealed heterogenous wasting phenotypes dependent on tumour genetics and location as well as dietary conditions [23,24,27]. Therefore, to identify some key features driving cachexia-like tissue wasting by tumours, we analysed the impact on muscle wasting of genetically distinct tumours in otherwise similar settings.

We generated different tumour types in larval imaginal discs in order to assess whether cachectic phenotypes could be observed in these animals and compared tumour-bearing and tumour-free animals. We quantified three phenotypic parameters associated with the human syndrome: tumour burden, cachexia-like muscle tissue wasting, and the presence of lipid droplets within muscles (Figure 1A). The latter was used as a parallel of the characteristic intramyocellular lipid droplet (IMLD) accumulation observed in human cancer cachexia [30]. Cachectic muscle wasting was quantified by assessing the percentage coverage of the cuticle by body wall muscle, tumour burden by assessing imaginal disc volume, and IMLDs by quantifying the frequency of lipid droplets per unit area of the 5th segment of the 7th ventral muscle (Figure 1A).

We induced hyperplastic wing disc tumours through expression of constitutively active Yorkie (Yki), *Yki^S168A^* using the *rotund-gal4* driver. To model non-invasive neoplastic tumours, we used homozygous *discs large* (*dlg*) mutant animals, *dlg^40.2^*. Invasive neoplastic tumours were produced either through the formation of eye disc-specific *scribbled* mutant clones (*scrib^1^*) expressing *Ras^V12^* (*Ras^V12^*, *scrib^1^*) or expression of *Ras^V12^* and *scrib^IR^* under the control of *hedgehog-Gal4* (*Ras^V12^*, *scrib^IR^*) (Figure 1B–F and Appendix A).

As previously demonstrated, neoplastic tumours induce developmental delay [31] (Appendix A). A mild muscle wasting phenotype was detected at 9 days after egg deposition (AED) in delayed animals with *Ras^V12^*, *scrib^1^* or *Ras^V12^*, *scrib^IR^* invasive neoplastic tumours as quantified by the coverage of the larval cuticle by body wall muscle (Figure 1K). At 12 days AED a strong muscle wasting phenotype could be observed in these animals, while larvae with non-invasive neoplastic *dlg^40.2^* tumours showed a milder, though significant, wasting phenotype (Figure 1G–K and Appendix A). The characteristic IMLD phenotype seen in the wasting muscle of cachectic patients was recapitulated in the wasting muscle of cachectic larvae, although this phenotype was much milder in *dlg^40.2^* tumour-bearing larvae at 12 days AED and showed a non-significant trend (Figure 1L–P and Appendix A). Developmental delay alone was not sufficient to induce wasting, as tumour-free developmentally delayed *ecdysoneless* mutant (*ecd^1^*) larvae showed neither muscle wasting nor IMLD formation (Figure 1K,P and Appendix A). Consistent with a previous report, hyperplastic *Yki^S168A^* imaginal disc tumours were not cachectogenic [24], and showed neither larval muscle wasting nor IMLD formation (Figure 1H,K,M-M″,P).

Our results suggest that the genetic characteristics of the tumour, which define the tumour phenotype as either neoplastic or hyperplastic, and invasive or non-invasive, determine the presence and degree of cachectic muscle wasting. However, *Yki^S168A^* tumours did not induce the strong developmental delay seen in neoplastic tumour genotypes (Appendix A). Even though *Yki^S168A^* tumours developed to a much larger size than any other tumour at 6 days AED (Figure 1F and Appendix A), wasting only becomes evident in the context of extensive developmental delay, and in tumours significantly larger than those of the oldest occurring *Yki^S168A^* larvae (Figure 1B–K and Appendix A). It was therefore conceivable that longer developmental timing in the context of *Yki^S168A^* tumour burden might ultimately lead to cachexia. Indeed, hyperactivation of *Yki* in a model of adult intestinal tumourigenesis leads to peripheral host tissue wasting [23]. To test such possibility, we generated hyperplastic *Yki^S168A^* tumours in the background of the *ecd^1^* mutation, in order to produce large hyperplastic tumours in the context of developmental delay. We compared these tumours to cachectogenic *Ras^V12^*, *scrib^1^* tumours of the same size (Figure 2A,B,G), and found that *Yki^S168A^*, *ecd^1^* animals displayed no cachectic phenotypes of muscle wasting (Figure 2C,D,H) or IMLD formation (Figure 2E–F″,I).

Altogether, these results demonstrate that tumour burden in the context of developmental delay was not sufficient to drive cachexia and that it is the genotype of the tumour, which is critical to the presentation of muscle wasting. This is consistent with observations in human patients, where specific cancer types such as gastric and pancreatic cancers have a very high prevalence of cachexia (85% and 83%, respectively) [13] and more aggressive tumours are associated with the presentation of wasting even when small in size [12,32,33].

### 2.2. Starvation and Liquid Retention Are Not Sufficient to Drive Cachexia

Anorexia is one of the main clinical manifestations of cachexia and can drive tissue wasting [4]. A recent adult eye tumour model in *Drosophila* revealed a conserved mechanism by which pro-cachectic tumours induce anorexia, pre-ceding host tissue wasting [26] However, anorexia is not considered a primary driver of cancer cachexia, as reversing it is not sufficient to fully rescue the loss of tissue [34,35]. During normal development, *Drosophila* larvae feed almost constantly up until the wandering L3 stage. Given larvae stop feeding at the end of the L3 stage and developmentally delayed larvae persist in this stage, we next considered the possibility that the muscle wasting seen in tumour-bearing larvae could be a result of prolonged starvation.

To test our hypothesis, we quantified the amount of food ingested in 30 min at different time points by control *w^1118^* and *ecd^1^* larvae, and larvae with *Yki^S168A^*; *dlg^40.2^*; *Ras^V12^*, *scrib^1^* or *Ras^V12^*, *scrib^IR^* tumours (Figure 3A). As expected, larval feeding was decreased by the late L3 stage at 6 days AED (Figure 3B). However, in developmentally delayed larvae, no differences in the levels of feeding were observed past 8 days AED, where all larvae analysed, including tumour-free developmentally delayed *ecd^1^* larvae that do not undergo muscle wasting, had negligible food intake (Figure 3B). Forced starvation of *ecd^1^* larvae was not sufficient to drive wasting either (data not shown). We next examined the feeding behaviour of *Yki^S168A^*, *ecd^1^* animals in the same manner, and found no differences in food intake between non-wasting *Yki^S168A^*, *ecd^1^* animals and cachectic larvae with *Ras^V12^*, *scrib^1^* tumours (Figure 3C). Altogether, these data strongly suggest that starvation is not sufficient to induce muscle wasting in the context tumour burden in these larval models. However, as our experimental design does not allow us to measure nutrient absorption by the intestine, we cannot fully discard the possibility that differences in nutrient absorption between cachectic and non-cachectic models could contribute to tissue wasting.

Abdominal fluid accumulation is often present in cachectic cancer patients [4]. Similarly, bloating is a phenotype commonly associated with tumour-bearing larvae. Bloating is essentially the result of fluid retention, which generates an oedema-like phenotype where the larval cuticle, and therefore the body wall muscle, becomes stretched. Developmentally delayed tumour-bearing larvae contain a much larger amount of haemolymph than wild-type larvae [19], which gives rise to the characteristic ‘giant larvae’ phenotype seen in these animals.

Bloating can be induced in larvae carrying transheterozygous mutations for a regulator of BMP signalling, *larval translucida* (*ltl)* [36]. We found that ubiquitous expression of an RNAi for *ltl* (*ltl^IR^*) using a *tubulin-Gal4* driver (*tub > Gal4*) recapitulated the bloating phenotype observed in *ltl* mutants, allowing the assessment of body wall muscle in tumour-free, bloated larvae (Figure 3D–L). To ensure the duration of the bloating time was consistent between genotypes, *ltl^IR^* overexpressing larvae were examined at seven days AED to match the time *Ras^V12^*, *scrib^1^* animals spent in the bloated stage by twelve days AED (Figure 3F–I). Past this seven-day time point *ltl^IR^* larvae became much more distended than tumour-bearing animals, and some lethality occurred (data not shown). We saw no cachectic phenotypes in *ltl^IR^* larvae (Figure 3M–S). These results suggest that, as in the case of starvation, the mechanical stress caused by liquid retention is not sufficient to drive the body wall muscle wasting observed in cachectic tumour-bearing animals.

### 2.3. Inflammation and JAK/STAT Signalling Is Elevated in Cachectic Larval Models

*Scrib^−/−^*, *Ras**^V12^* imaginal disc tumours induce cachexia when transplanted into adult hosts through ImpL2 secretion [24]. We therefore tested the involvement of *impL2* in *scrib^IR^*, *Ras**^V12^* larval tumours driven by *hh-gal4* (Figure 1F,K and Appendix A). Even though overexpression of *impL2* RNAi in this setting caused strong reduction in *impL2* in the tumour (Appendix A), we did not observe suppression of skeletal muscle wasting (Appendix A). A similar result was reported in a larval tumour model of diet-induced cancer cachexia [27].

To investigate which factors might be driving the cachectic wasting observed in our larval models we performed RNA sequencing (RNAseq) on cuticle and muscle of wild type and *Ras^V12^*, *scrib^1^* tumour-bearing larvae (Figure 4). Pathway and gene ontology analysis revealed significant deregulation of stress and starvation response genes, metabolic genes, and inflammation and immune regulatory genes within cachectic muscle of tumour bearing larvae (Figure 4A–C and Appendix A). Consistently with the developmental arrest displayed by tumour bearing larvae, we observed a reduced expression of genes involved in ecdysteroid metabolism and response. We also observed a dramatic decrease in several metabolic processes involved in critical cellular functions. Importantly, genes related to mitochondrial ATP synthesis, glucose metabolism and amino acid synthesis were downregulated, recapitulating key features of the human condition as well as other cachectic models [37,38,39,40,41].

RNAseq analysis of cuticles and wasting muscles also revealed upregulation of pathways involved in protein degradation, the response to starvation and oxidative stress (Figure 4B,C). Trypsin-like cysteine/serine proteases, suggested to be mediators of autophagy, were strongly upregulated in wasting muscles (Figure 4C). In addition, glutathione metabolism and disulphide bond formation were both increased suggesting that during wasting, muscles have to face a strong oxidative stress. Ubiquitin driven protein degradation has also been reported to be a driver of muscle wasting in both sarcopenia and cancer-induced cachexia [42]. Amongst the proteins involved in this process, the F-BOX ubiquitin ligase family member Atrogin-1 seems to be an important player in this muscle degradation process [43]. Interestingly, the closest fly *Atrogin-1* homologous gene, *CG11658*, was upregulated in our RNAseq analysis (Appendix A) suggesting a potential further analogy with the human condition. Finally, conserved signalling pathways mediating the cellular response to inflammatory cytokines were increased in wasting muscles, including multiple targets of the Janus Kinase/Signal Transducer and Activator of Transcription (JAK/STAT) signalling pathway (Figure 4D). This matches well with mammalian studies that have implicated JAK/STAT signalling in cachectic muscle wasting, either through the autonomous action of JAK/STAT signalling in the muscle [44,45,46] or the systemic action of the JAK/STAT pathway ligand IL-6 [39,47,48,49,50,51].

TNF-α was originally known as the ‘cachectic hormone’. Systemic increase in TNF-α and IL-6 are hallmarks of cancer cachexia [1]. However, anti-TNF-α therapies have been used for the treatment of cancer cachexia without much success [52,53]. Interestingly, the gene encoding for Wengen (*wgn*), a *Drosophila* TNF-α/Egr receptor [54,55] and downstream JNK singling were also upregulated in wasting muscles (Appendix A). Altogether, these data suggest that cachectic larval tumour models recapitulate multiple cellular and molecular manifestations of the human syndrome. In particular, the conserved activation of JAK/STAT and TNF-α signalling in cachectic muscles recapitulates the ‘high inflammation’ hallmark of cancer cachexia [51,56,57].

### 2.4. Establishing a Dual-Driver System to Study Tumour/Host Interactions in Cancer-Induced Cachexia

We next set up a *Drosophila* larval model carrying a dual driver system (DDS) that would allow concomitant, yet independent, genetic modifications of tumour and peripheral tissues. To achieve this, we used the *Neurospora crassa* derived QF/QUAS/QS system [58] to generate *QUAS-RFP* labelled control (Figure 5A,A′) or *scrib^1^*; *QUAS-**Ras^V12^* MARCM clones (Figure 5B,B′) within the larval eye/antennal disc using the imaginal disc driver *ET40-QF,* while sparing the *Gal4/UAS* system [59] to express transgenes of interest in the body wall muscles using *mhc-gal4* (Figure 5A,B′). Consistent with our findings in the conventional *scrib^1^*; *Ras^V12^* tumour model (Figure 1), tumours generated using our DDS system led to sever hyperplasia of the eye imaginal discs (Figure 5C), larval developmental delay and muscle wasting (Figure 5D). Furthermore, molecular assessment by RT-qPCR confirmed upregulation of the JAK/STAT signalling target *Socs36E*, *wgn* and *CG11658/Atrogin-1* expression in muscles from animals with DDS generated *scrib^1^*; *Ras^V12^* clones (Figure 6A–C).

### 2.5. Muscle Autonomous JAK/STAT and TNF-α Signalling Drive Tumour-Induced Muscle Tissue Wasting in Drosophila

Given the recognised importance of TNF-α and JAK/STAT signalling in human cancer cachexia [1], as a proof-of-principle, we used our DDS system to assess the functional role of these pathways in muscle wasting upon tumour bearing. To achieve this, we overexpressed RNA interference (RNAi) for *stat* and *wgn*, in the muscle of *scrib^1^*; *Ras^V12^* tumour bearing larvae. Knocking down either gene impaired progressive muscle wasting in tumour bearing animals (Figure 6D–K) suggesting that activation of JAK/STAT and TNF-α signalling in skeletal muscle of tumour bearing larvae contributes to muscle wasting induced by tumour in this model. Future studies would explore the origin of the signals activating JAK/STAT and TNF-α signalling in cachectic muscles, a potential interconnection between these two inflammatory pathways and the downstream molecular mechanisms through which they promote muscle degradation.

Altogether, we present previously uncharacterized models of tumour induced cachexia in developing *Drosophila* and generate a paradigm to investigate tumour/skeletal muscle crosstalk during cachexia, which may be also adapted for additional studies on tumour/host interactions in vivo.

## 3. Materials and Methods

### 3.1. Drosophila Stocks

A list of all fly stocks and the full genotypes of experimental crosses presented here can be found in Appendix A.

Dual driver system: The *QUAS-Ras^V12^* fly line has been generated for this study. Whole fly genomic DNA was extracted from flies carrying a *UAS-Ras^V12^* transgene, using the Omega Ezna Insect DNA kit (Omega Bio-tek, Inc., Norcross, GA, USA). *Ras^V12^* was PCR amplified using primers containing EcoRI and XbaII restriction sites (forward: *AGCGGATCCATGACGGAATACAAACTGGTC* reverse: *ACCTCTAGATTAGAGCATTTTACATTTAAATCTACG*, respectively). The amplified *Ras^V12^* PCR product was digested with the corresponding restriction enzymes and cloned into a *pQUAST* vector (Addgene number 24349; Addgene, Watertown, MA, USA) using standard experimental procedures described in [58]. The *pQUAST-Ras^V12^* construct was sent for injection (Rainbow Transgenic Flies, Inc., Camarillo, CA, USA). Transgenic flies carrying *QUAS-Ras^V12^* on the third chromosome were used for experiments.

### 3.2. Fly Husbandry and Genetics

All crosses were kept in temperature-controlled incubators with a 12 h light/dark cycle and raised on standard fly medium. Crosses were generally performed at 25 °C, apart from *ecd^1^* larvae and *Ras^V12^*, *scrib^1^* and *Yki^S168A^*, *ecd^1^* larvae, which were maintained at 22 °C for 4 days AED, and then shifted to 29 °C to induce *ecd^1^*-dependent developmental delay.

### 3.3. Immunofluorescent Staining

Cuticles and tumours were dissected on Sylgard plates in 1X PBS, and subsequently fixed on the plate in 4% formaldehyde (Polysciences, Inc., Oak Ridge, TN, USA) for 30–40 min. After fixation, tissues were transferred to a nine-well glass dissection plate for three wash steps of 15 min, on an orbital shaker at 80 rpm.

Cuticles were washed and stained in 0.05% Saponin in PBS (PBSS), while tumours were washed and stained in PBS containing 0.1% Triton X-100 (PBST). Cuticles were stained overnight at 4 °C with DAPI, Phalloidin-488 (Invitrogen, Waltham, MA, USA) at 1:100, and LipidTOX Deep Red (Thermofisher Scientific, Waltham, MA, USA) at 1:500. Cuticles were washed 3 times in PBSS before mounting on glass slides in Vectashield mounting media without DAPI (Vector Laboratories, Inc., San Francisco, CA, USA). Tumours were stained overnight at 4 °C with DAPI, washed three times in PBST and mounted on glass slides with a spacer in Vectashield mounting media without DAPI.

### 3.4. Imaging and Image Processing

All immunofluorescent images were captured on a Zeiss 710 confocal microscope system (Carl Zeiss AG, Jena, Germany). Brightfield images were captured on a Leica M205 FA microscope with Leica DFC 500 camera (Leica, Wetzlar, Germany). Image processing was carried out in ImageJ (NIH, Bethesda, MD, USA) and figure panels were created in Adobe Photoshop CS5 (Adobe, San Jose, CA, USA).

### 3.5. QPCR

RT-qPCRs were done on 2–4 biological replicates. Each sample/biological replicate was prepared with tissue from 5–10 larvae. Samples were homogenised using a pestle and mortar, and RNA was extracted using QIAGEN RNA extraction columns with DNAse treatment (Qiagen, Hilden, Germany). cDNA conversion was performed using the High-Capacity cDNA reverse transcription kit (Applied Biosystems, Waltham, MA, USA). SYBR Green FastMix Low ROX (Quanta Bio, Plain City, OH, USA) was used for qPCRs. The expression of target genes was assessed relative to the expression of the *rpl32* housekeeping gene, with a series of 10-fold sample dilutions used to produce a standard curve to quantify expression. qPCR was performed on Applied Biosystems 7500 Fast Real Time PCR machine (Applied Biosystems, Waltham, MA, USA) and analysed using the system software. Melt curves were generated for the first use of each primer pair to ensure no primer-dimer or multiple product formation occurred. Primer pairs used can be found in Appendix A.

### 3.6. Quantification of the Percentage of Body Wall Muscle Covering the Cuticle

To quantify systemic body wall muscle wasting in larvae the following method was developed, using ImageJ software. LSM Z-stack tile-scans of whole cuticles taken at 10X were analysed. Images underwent maximal Z-projection, channels were merged into a composite image, and then the composite was stacked to an RGB image. The outline of the cuticle was then manually delineated, and the area of this selection was measured in pixels, to provide the value for the area of the whole cuticle. The area outside this selection was then cleared, to ensure that no signal outside of the specific area of interest was quantified. Actin in muscle was stained with phalloidin, with signal intensity presented in ImageJ’s standard green look-up table. The Colour Threshold tool was then used to detect the green pixels that represented the actin in muscle. The Hue slider was set to include only the green region of the colour spectrum, and the Brightness slider was adjusted to alter the threshold call to remove any background staining. Once muscles were highlighted, this area was selected and measured in pixels. The percentage coverage of the cuticle by muscle was then calculated from these figures by dividing the value for total pixel area of muscle by the value for total pixel area of the whole cuticle and multiplying by 100.

### 3.7. Quantification of Lipid Droplet Frequency in Ventral Muscle

To provide a quantitative measure of the lipid droplet phenotype in wasting muscle the following method of quantifying the frequency of lipid droplets observed in muscle was developed, using the ImageJ software. LSM Z-stacks of the 5th segment of the 7th ventral muscle underwent maximal Z-projection, and the channels were merged to form a composite image. The outline of the muscle was manually delineated, and the outline saved to the Region of Interest (ROI) manager. The area of this selection was then measured in pixels and recorded. The channel for LipidTOX dye that stained lipid droplets was then isolated, and the channel converted to an RGB image. The ROI manager of the muscle outline was used to highlight the region of the image previously selected, and all areas of the image outside of this region were excluded. The number of lipid droplets in the muscle was then calculated using the Analyse Particles tool, with settings of Size: 1–3 μm, and Circularity: 0.30–1.00. The number of lipid droplets was then divided by the area of muscle in pixels, to provide a comparable measure of the frequency of lipid droplets per unit area of muscle.

### 3.8. Quantification of Tumour Volume

The Volocity 3D image analysis software (Volocity 6.3.0, Quantitation [Analysis 2D, 3D, 4D] + Base Package) (Quorum Technologies Inc., Puslinch, Ontario, Canada) was used in the quantification of tumour volume. Tumour images were cropped to include only the individual tumour, and the software was used to calculate tumour volume. Tumour volume was determined based on the quantification of the total volume of DAPI signal in the LSM Z-stack. The ‘Fill Holes in Object’ plugin was utilised to ensure that the small spaces between nuclei were included in the quantification for a more accurate assessment of total tumour volume. This quantification process produced a raw data table listing the volume of each surface identified in the image of the tumour. The total volume was then calculated by identifying the sum of all the surfaces identified in each tumour image, using the Volocity results Analysis package.

### 3.9. Larval Feeding Assessment

Fly food made in the Beatson Central Services unit was melted in a microwave. Food was left to cool until hand-hot, and Erioglaucine Disodium salt (Sigma, St. Louis, MO, USA) was added at a concentration of 0.01 g/mL. Larvae were collected and washed with distilled water in a nine-well glass dissection plate, then added to the coloured food for 30 min. Larvae were removed and washed to remove traces of coloured food. Single larvae were then placed in 1.5 mL Eppendorf tubes (Eppendorf, Hamburg, Germany) and snap frozen on dry ice, ready for quantification. Frozen samples were lysed in 100 μL distilled water using a pestle and spun at 4 °C at 10,000 rpm for seven minutes to remove tissue fragments. The supernatant was then collected and 50 μL was added to a 96-well tissue culture plate (Falcon, Lagos, Nigeria). Dye standards at 0.000, 0.003, 0.006, 0.012, 0.025 and 0.050 μg/μL were also added to generate a standard curve for dye concentration. The intensity of the dye was analyzed using a Tecan Sunrise plate-reader (Tecan AG, Switzerland), with OD absorption set at 630 nm. The OD value of blank wells was subtracted from all standards, and these normalized values were used to generate a standard curve. OD values for samples were similarly normalized, and the standard curve was then used to calculate the concentration of the dye in each well. This dye concentration was then multiplied by the volume of the buffer the larva was lysed in (100 μL) to provide an absolute value for the mass of the dye ingested by each individual larva.

### 3.10. RNA Sequencing

Individual larval cuticle and muscle samples were dissected as previously described, including the removal of the anterior of the animal with microscissors to exclude the mouthparts and spiracles. As soon as viscera were removed cuticles were frozen in a collection Eppendorf tubes placed on dry ice.

RNA from 30 cuticles plus skeletal muscle per sample was extracted using QIAGEN RNA extraction kits with DNAse treatment, with tissue dissociated using a pestle and mortar and then by a QIAshredder column (QIAGEN). RNA quality was assesed on a Bioanalyzer, and quantity was estimated using a NanoDrop spectrophotometer (Thermofisher Scientific, Waltham, MA, USA).

Suitable RNA samples were passed to the Beatson Molecular Technology Service, who performed cDNA library preparation and ran the RNAseq on the Illumina GAIIx sequencer (Illumina, In., San Diego, CA, USA). Raw data were processed by the Beatson Institute Computational Biology group, who analysed the data to generate tables of all genes detected in the RNAseq, and the Log2 fold change of gene expression in the cuticles of *Ras^V12^*, *scrib^1^* tumour-bearing larvae as opposed to the cuticles from the *w^1118^* control. Heatmaps were generated using the MeV software. The read counts of genes of interest were mean-centred to provide a comparable measure of up- or downregulation of gene expression in the cuticles of *Ras^V12^*, *scrib^1^* tumour-bearing, or *w^1118^* control larvae. The table containing full RNAseq data can be found in Appendix A.

### 3.11. Statistics and Reproducibility

Graphs were generated using GraphPad Prism 6 (GraphPad Software, San Diego, CA, USA). *t* test was for two-group comparisons and one-way ANOVA with Tukey or Dunnett post hoc corrections was used for comparison of multiple samples. Relevant p-values are included in figure legends. With the exception of the data in Figure 6K and Appendix A all experiments were repeated 2 or 3 times and contain tissue samples from at least 10 animals (*n*) per repeat. Data in Figure 6K and Appendix A corresponds to biological duplicates with a total of 8–13 animals. Each dot in dot plot graphs correspond to an individual sample (*n*). Mean ± SEM is indicated in each graph.

## Figures and Tables

**Figure 1 ijms-22-08317-f001:**
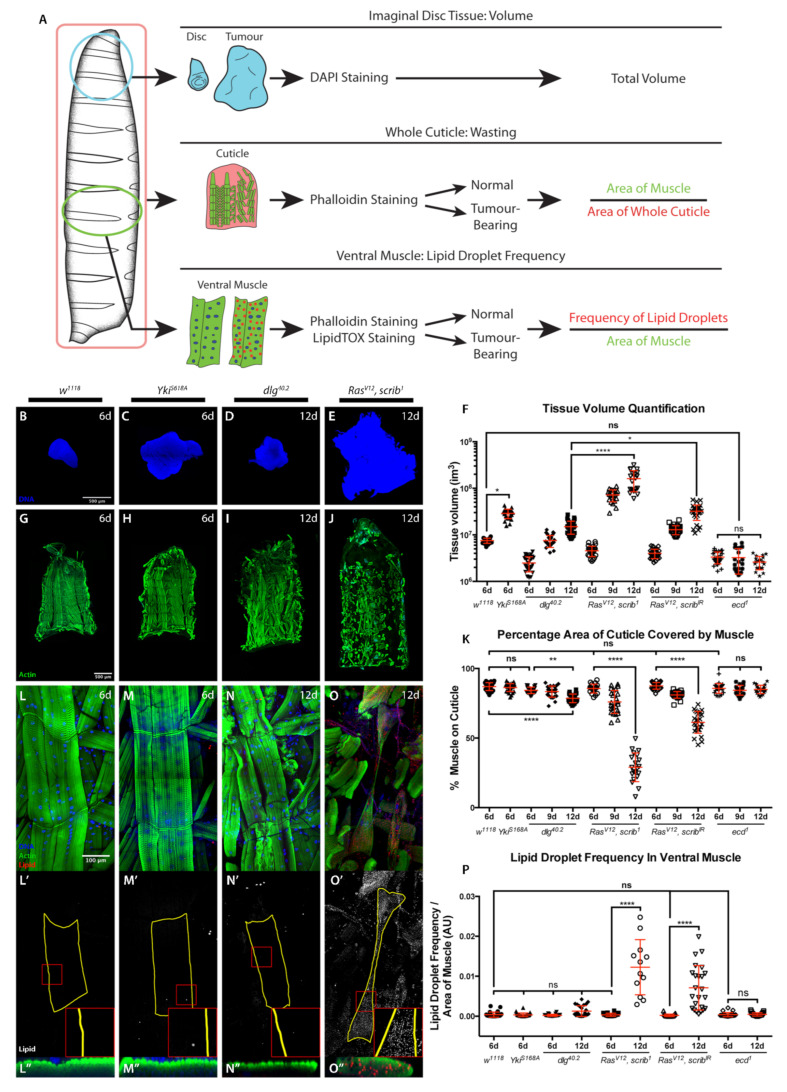
**Cachectic muscle wasting is dependent on tumour genotype.** (**A**) Schematic of experimental approach used to assess tumour volume, muscle wasting and IMLDs. (**B**–**E**) Imaginal tissues stained with DAPI. Normal wing disc at 6 days AED (*w^1118^*) (**B**). Hyperplastic tumour at 6 days AED (*rotund* > *Yki^S168A^*) (**C**), non-invasive neoplastic tumour at 12 days AED (*dlg^40.2^*) (**D**) and MARCM-generated eye invasive neoplastic tumour at 12 days AED (*Ras^V12^*, *scrib^1^*) (**E**). (**F**) Tissue volume quantification. (**G**–**J**) Cuticles with muscle actin stained with Phalloidin (green) from animals of genotypes and ages as in (**B**–**E**). (**K**) Quantification of the percentage of cuticle covered by muscle. (**L**–**O″**) 5th segment of the 7th ventral muscle, stained with DAPI (blue), Phalloidin (green), and LipidTOX to visualize lipids (red or grey) in body wall muscles from animals of genotypes and ages as in (**B**–**E**). All panels denoted (′) show lipid staining only. All panels denoted (“) show a transversal orthographic view of muscle with a 2X zoom, stained with Phalloidin and LipidTOX. (**P**) Quantification of the frequency of IMLDs in the 5th segment of the 7th ventral muscle, expressed as a ratio over the area of that muscle. All graphs show one-way ANOVA with Tukey post hoc correction. Mean ± SEM is indicated. ns: non-significant, *p* > 0.05; * *p* ≤ 0.05; ** *p* ≤ 0.01; **** *p* ≤ 0.0001.

**Figure 2 ijms-22-08317-f002:**
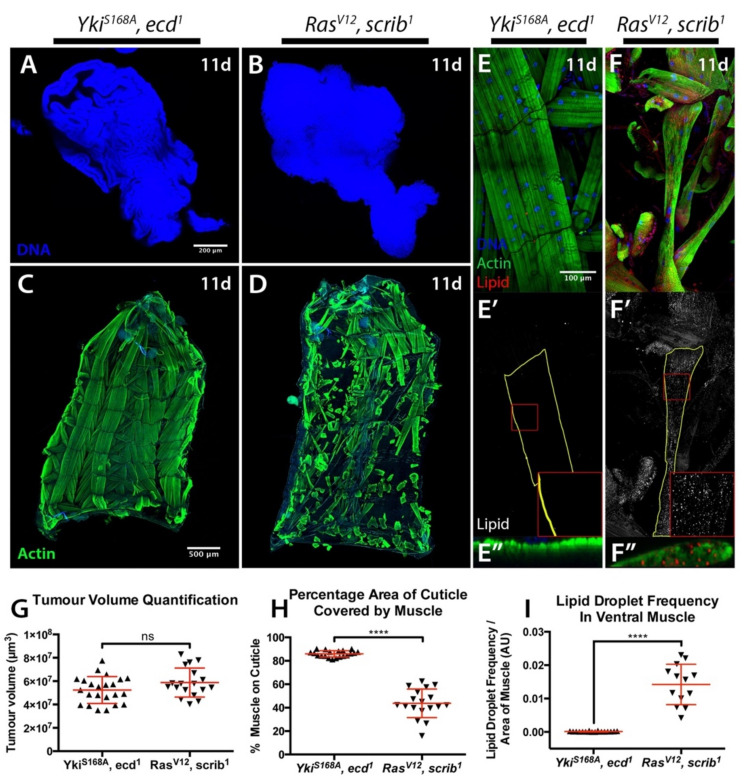
**Tumour size is not sufficient to drive cachexia.** (**A**,**B**) Tumours stained with DAPI. *rotund > Yki^S168A^* tumours induced in an *ecd^1^* mutant background (**A**), and MARCM eye *Ras^V12^*, *scrib^1^* tumours (**B**), both at 11 days AED. (**C**,**D**) Cuticles from animals as in (**A**) and (**B**), with muscle actin stained with Phalloidin (green). (**E**–**F**″) 5th segment of the 7th ventral muscle, stained with DAPI (blue), Phalloidin (green), and LipidTOX (red or grey). Animal ages and genotypes as in (**A**) and (**B**). Panels denoted (′) show lipid staining only. Panels denoted (″) show a transversal orthographic view of muscle with a 2X zoom. (**G**) Tissue volume quantification. (**H**) Quantification of the percentage of the cuticle covered by muscle. (**I**) Quantification of the frequency of IMLDs in the 5th segment of the 7th ventral muscle, expressed as a ratio over the area of that muscle. All graphs show *t* test. Mean ± SEM is indicated. ns: non-significant, *p* > 0.05; **** *p* ≤ 0.0001.

**Figure 3 ijms-22-08317-f003:**
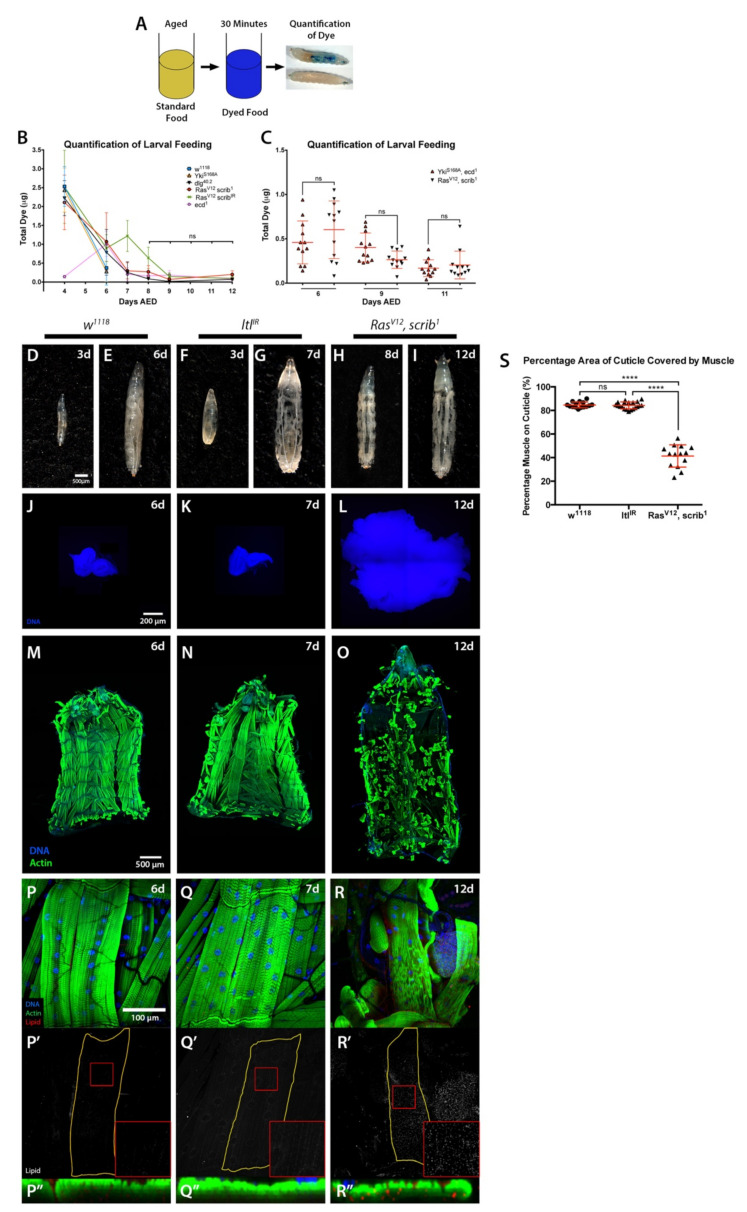
**Starvation or bloating are not sufficient to drive cachexia.** (**A**) Schematic of the approach for assessment of larval feeding. (**B**,**C**) Quantification of dye levels consumed by larvae of the indicated genotypes. (**D**–**I**) Images of whole larvae. Normal (*w^1118^*) larvae at 3 (**D**) and 6 (**E**) days AED. Tumour-free bloated larvae with ubiquitous *tubulin > Gal4*-driven expression of *ltl^IR^* at 3 (**F**) and 7 (**G**) days AED. *Ras^V12^*, *scrib^1^* tumour-bearing larvae at 8 (**H**) and 12 (**I**) days AED. (**J**–**L**) Imaginal tissues stained with DAPI. Normal eye-antennal disc at 6 days AED (*w^1118^*) (**J**), eye-antennal disc at 7 days AED (*tub > ltl^IR^*) (**K**), and invasive neoplastic eye tumours at 12 days AED (*Ras^V12^*, *scrib^1^*) (**L**). (**M**–**O**) Cuticles with muscle actin stained with Phalloidin, animals of ages and genotypes as described for (**J**–**L**). (**P**–**R″**) 5th segment of the 7th ventral muscle, stained with DAPI (blue), Phalloidin (green), and LipidTOX (grey or red). Animal ages and genotypes as described for (**J**–**L**). All panels denoted (‘) show lipid staining only. All panels denoted (“) show a transversal orthographic view of muscle with a 2X zoom. (**S**) Quantification of the percentage of the cuticle covered by muscle. All graphs show one-way ANOVA with Tukey post hoc correction. Mean ± SEM is indicated. ns: non-significant, *p* > 0.05; **** *p* ≤ 0.0001.

**Figure 4 ijms-22-08317-f004:**
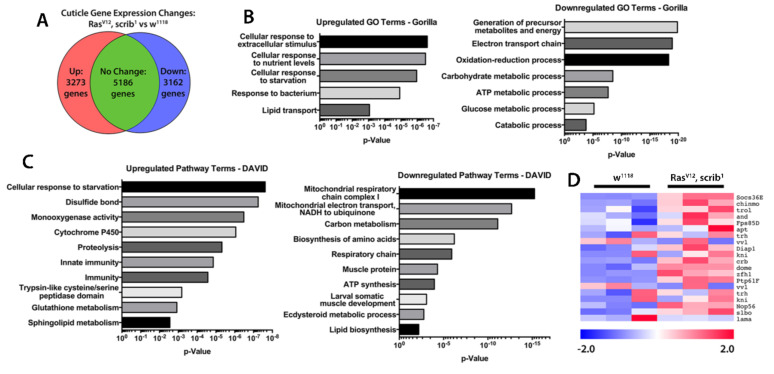
**Molecular characterization of cachectic muscles from tumour bearing larvae.** (**A**) Frequency of up- and downregulated expression of genes identified in an RNAseq from cuticles and muscle of normal (w^1118^) and MARCM Ras^V12^, scrib^1^ tumour-bearing animals. (**B**) List of GO terms enriched in up- and downregulated gene sets from RNAseq. (**C**) List of pathway terms enriched in up- and downregulated gene sets from RNAseq. (**D**) Heatmap of relative expression levels of JAK/STAT pathway genes from RNAseq. Heatmap colour intensity displays log_2_-fold change values relative to the mean gene expression level across both genotypes.

**Figure 5 ijms-22-08317-f005:**
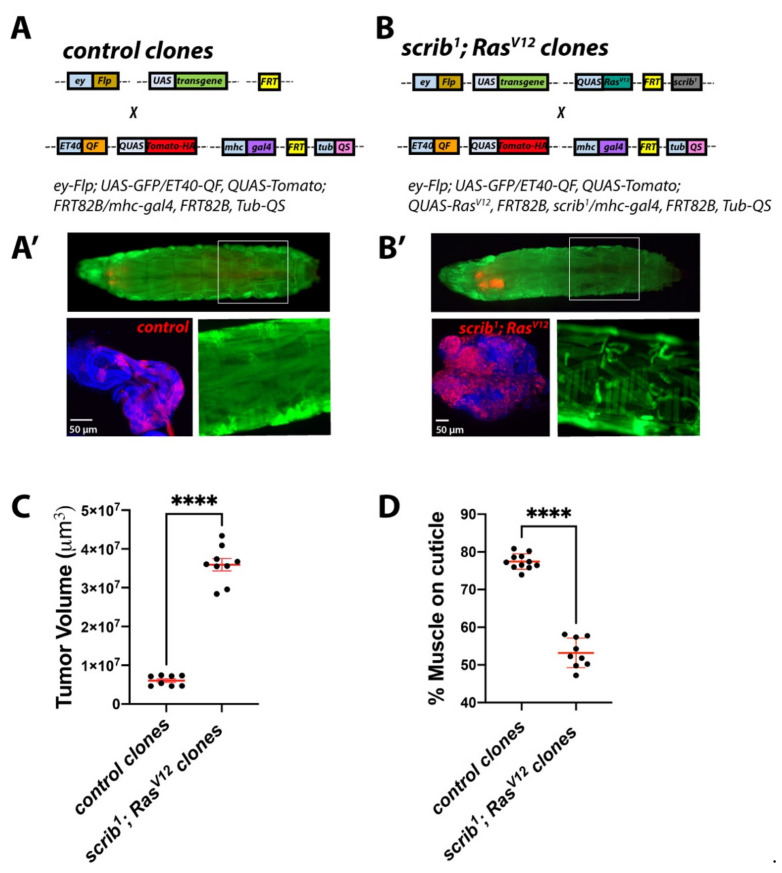
**Generation of a dual driver system for independent genetic manipulation of skeletal muscle and larval imaginal discs.** (**A**,**B**) Schematic of the genetic components of a dual driver system used for genetic manipulation of skeletal muscle in larvae bearing eye/antennal disc control or *Ras^V12^*, *scrib^1^* clones. The imaginal disc driver *ET40-QF* was used to drive *QUAS-tomato* with and without *QUAS-Ras^V12^* for the generation of *Srcib*; *Ras^V12^* MARCM clones and *mhc-gal4* was used for *UAS-transgene* expression in the skeletal muscle. (**A′**,**B′**) Larvae with dual driver system generated control (**A′**) or *scrib^1^*; *Ras^V12^* clones (**B′**) (red; in larvae and bottom left panels) and muscle specific concomitant overexpression of *UAS-gfp* (green; in larvae and bottom right panels). Boxed areas within larval images delineate magnified views of muscle (bottom right panels). Full genotypes are indicated on top of immunofluorescence images. 6-day-old control (**A′**) and 10-day-old *scrib^1^*; *Ras^V12^* larvae (**B′**). (**C**,**D**) Quantification of tissue volume (**C**) and percentage of the cuticle covered by muscle (**D**) in animals of ages and genotypes as in (**A′**,**B′**). All graphs show *t* test. Mean ± SEM is indicated. **** *p* ≤ 0.0001.

**Figure 6 ijms-22-08317-f006:**
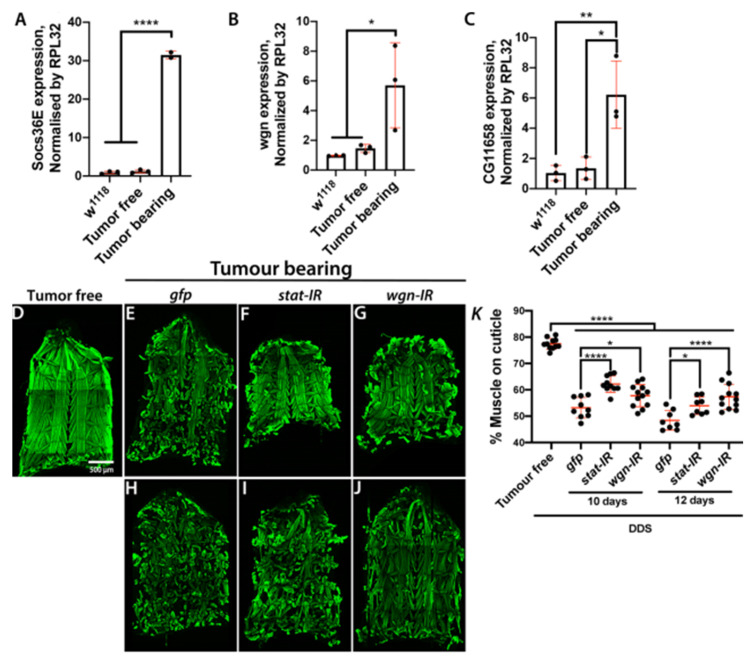
**Muscle JAK/STAT and TNF-α signalling drives tissue wasting.** (**A**–**C**) mRNA expression levels of the JAK/STAT pathway target *Socs36E* (**A**), the TNF-α/Egr receptor *wgn* (**B**) and the *Atrogin 1* homologue *CG116758* (**C**) in skeletal muscle from 6-day old wild type larvae (*w^1118^*) or larvae with dual driver system (DDS) generated control (Tumour free) or *scrib^1^*; *Ras^V12^* clones (Tumour bearing). (**D**–**J**) Cuticles from larvae with DDS generated control clones (Tumour free; 6 day-old) (**D**) or *scrib^1^*; *Ras^V12^* clones (Tumour bearing) with muscle specific concomitant overexpression of a control transgene (*gfp*), *stat RNAi* (*stat-IR*) or *wgn RNAi* (*wgn-IR*) (**E**–**J**). Cuticles of tumour bearing animals were assessed at 10 days (**E**–**G**) and 12 days (**H**–**J**) of larval age (AED). Muscle actin was stained with Phalloidin (green). (**K**) Quantification of the percentage of cuticle covered by muscle in larvae of genotypes and ages as in (**D**–**J**). All graphs show one-way ANOVA with Tukey post hoc correction. Mean ± SEM is indicated. * *p* ≤ 0.05; ** *p* ≤ 0.01; **** *p* ≤ 0.0001. DDS: dual driver system.

## Data Availability

Reagents and any necessary information related to this study will be available from the corresponding author upon request. RNA sequencing data including all raw sequence files and processed files have been deposited in the Gene Expression Omnibus under accession number GSE178332 and can be accessed through the following link: https://www.ncbi.nlm.nih.gov/geo/query/acc.cgi?acc=GSE178332 (accessed on 25th July 2021).

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
