# Peer review of "Drosophila Larval Models of Invasive Tumorigenesis for In Vivo Studies on Tumour/Peripheral Host Tissue Interactions during Cancer Cachexia"

_ijms, 2021, doi:10.3390/ijms22158317_

Round 1

Reviewer 1 Report

[IJMS] Manuscript ID: ijms-1296729

Hodgson et al

The manuscript by Hodgson et al describes the use of the Drosophila system to investigate the mechanisms and causes of cancer cachexia. The authors generate tumors in fly larvae, or induce other systemic effects, and then assay how these changes impact muscle atrophy. The authors generate clear data demonstrating that tumors arising from different genetic origins may differ in their ability to trigger atrophy, and also show that downstream tumor effects, such as liquid retention, do not in and of themselves cause atrophy. The authors also carry out RNAseq analysis of cachectic larval tissue and identify significant similarities with mammals in the genes and pathways that are affected, including JAK/STAT signaling and TNF1 signaling. Finally, the authors develop a genetic system that enables them to induce tumors while at the same time manipulating other aspects of gene expression, and show that down-regulation of one factor in each of the JAK/STAT or TNF1 signaling pathways can rescue the muscle atrophy phenotype arising from tumors.

Overall, I liked this paper very much. It was interesting and for the most part clearly described; the data are significant and appropriate controls and statistical tests are in place; and the findings are relevant to human medicine. I have a few minor issues that I would like the authors to address, as listed below.

  1. Figure 1B shows a control wing imaginal disc, however panel E shows a mutant eye disc. Please could the authors add an image of a control eye disc to panel 1B?

  1. There are a few items related to Figure 3 that should be addressed: i. The authors demonstrate that larvae with tumors eat a similar amount to larvae without tumors, and therefore conclude that the cachexia is not due to malnutrition. While this is likely to be the case, the authors cannot exclude the possibility that systemic defects affects nutrient uptake from the gut. An analysis of larval hemolymph could address this alternative interpretation, and I would suggest the authors either carry out this experiment or modify their conclusions. ii. Panel O’ should be Q’. iii. The legend describes panel T, but there is no panel T in the figure. iv. Results are described as being not significant if p is greater than or equal to 0.05, but is considered significant if p is less than or equal to 0.05. Based upon this description, a p value of 0.05 would be both significant and not significant. Sorry to be pedantic, but this should probably be addressed.

  1. Figure 4 and the description of it in the text should be modified to make it clear that the RNAseq analysis is not just of the muscle but muscle plus cuticle.

  1. The genetic scheme in Figure 5 is not clear in the figure and is not well described in the text. The genotypes of control and mutants should be indicated, and the QUAS system should be described in more detail for the non-Drosophila reader. In addition, panel B would benefit from images of a control larvae alongside the one containing the tumor.

Author Response

Please see attachment with rebuttal letter.

Reviewer 2 Report

A very detailed and well presented paper on the use of Drosophila as a model for cancer cachexia in human patients. The comparison of several types of tumors, several types of cachexia measures and good graphics for showing the comparisons make this a very strong paper of making this model available for those working on systems closer to human. The actual value of this for patients will remain to be demonstrated, but this work has much to offer in showing the presence of metabolic responses in muscle and fat to factors from tumors, and the appetite and bloating work increases this as a well described new model. The next phase would be to find the mediators of the cachexia in this system.

In fact, one might be concerned that the mediators have not been studied while jumping to the conclusion of inflammatory mediators in STAT and TNF, which is largely speculative and needs the dual driver system to provide some possibly relevant activity in the cachectic models. However, the comparative structure of the study, with several types of tumors being included makes even these less convincing parts still of interest.

One thing that might be added in discussion is that the delay of development has a strong similarity to the inhibition of differentiation seen in mammalian muscle by a number of factors most notably those such as TNF-alpha, mediated by the NF-kB signaling pathway. This observation has received much attention in the muscle literature even in cachexia. My personal bias is that these phenomena, though easily studied and quite observable, possibly are not so important in the human case of cancer cachexia in which the patients are mostly aged and for whom new muscle is not so much an issue as much as trying to inhibit the loss of fully differentiated muscle mass in the disease.

Author Response

(The authors gave the same response as above.)
